# REGRET ANALYSIS OF HYBRID LINEAR BANDITS WITH BIASED OFFLINE DATA

## ABSTRACT

Linear bandits have been extensively studied due to their broad applications and solid theoretical foundations. However, purely online algorithms often suffer from high exploration costs, while purely offline approaches critically depend on the quality of offline data. To bridge this gap, we study a hybrid setting where (biased) offline data is available in online learning. We propose Hybrid LinUCB, an algorithm that leverages both offline and online information by constructing two confidence ellipsoids to trade off bias against the size of offline data. We establish an upper bound and a nearly matching lower bound that explicitly capture the dependence on the bias upper bound $V$ and the spectrum of the offline feature matrix $V_0$. Compared with existing work, our algorithm requires weaker assumptions on offline data and exhibits stronger adaptability. Moreover, our theoretical analysis recovers and unifies prior guarantees across different settings.

## 1 INTRODUCTION

The bandit learning problem formulates a sequential decision process where an agent repeatedly interacts with an unknown environment (Lattimore & Szepesvári, 2020; Agrawal & Goyal, 2012). At each time step, the agent submits an action and receives noisy feedback from the environment. A key challenge in bandit learning is the exploration–exploitation trade-off: the agent must balance exploiting actions known to generate high rewards with exploring uncertain alternatives that may be superior. This bandit learning model has been successfully applied in many real-world domains, such as recommendation systems (Li et al., 2010), computational advertising (Geng et al., 2021), and more (Lu et al., 2021; Wang et al., 2025).

In classical bandit learning, the agent begins with no historical data, necessitating extensive exploration to identify the optimal action — a requirement that is often impractical in many real-world scenarios (Kapp, 2006). However, before the online learning begins, the agent may access previously collected offline logs that contain data relevant to the unknown environment (Li et al., 2022; Oetomo et al., 2023). Given that, a natural solution is to leverage offline datasets to reduce the reliance on online exploration, thereby enabling the faster identification of optimal actions. Numerous works have attempted to devise offline-to-online bandit learning methods for reducing the regret suffered in the online learning stage (Shivaswamy & Joachims, 2012; Oetomo et al., 2023; Zhang et al., 2025; Tan & Xu, 2024). Some recent works like warm-started LinUCB and warm-started Thompson Sampling, leveraging offline datasets to accelerate online exploration in linear bandits (Oetomo et al., 2023). There is another line of work called hybrid Reinforcement Learning (RL) focusing on leveraging offline datasets to enhance online exploration in Markov Decision Process (MDP) (Tan & Xu, 2024; Song et al., 2023).

However, most of these works assume that offline and online data follow the same distribution, which is too strong in reality due to the widespread nature of distribution shifts (Qu et al., 2025). Recently, a MIN-UCB algorithm is proposed by Cheung & Lyu (2024) to address stochastic MAB with possibly biased offline datasets. In parallel, for the tabular MDP setting, Qu et al. (2025) investigate a hybrid transfer RL problem where the offline MDP shifts from the online one, and develop the HySRL method to tackle this challenge. Nevertheless, these works primarily focus on the *arm-level* (MAB) or *state-action level* (tabular MDP) structures. In contrast, *linear bandits inherently emphasize the geometry of the entire action space*, where arm-level analyses and techniques cannot be directly applied. For instance, the proof strategies for LinUCB (linear bandits) versus UCB (MAB) differ

fundamentally, not only in regret upper bounds but also in lower bound constructions (Lattimore & Szepesvári, 2020). As a result, existing hybrid methods provide only *high-level inspirations* (e.g., order-wise comparisons, intuitive use of offline data), but not concrete techniques or transferable analyses for the linear bandit setting.

Moreover, as the number of arms grows, the benefits that such arm-level algorithms or analyses bring to online learning become increasingly limited; in contrast, linear bandits directly exploit the underlying structure of the entire feature space and thus do not suffer from this scalability issue. A simple example illustrates this gap: in MAB, having offline data on a subset of arms essentially reduces the number of online pulls needed for those arms. By contrast, in linear bandits, offline samples from certain arms inform the *entire feature space*, enabling more efficient global exploration rather than merely saving pulls on individual arms. This further highlights that the linear bandit problem requires a fundamentally different analytical perspective, distinct from existing works.

Based on these observations, we raise the following natural question:

*Can we design efficient linear bandit algorithms capable of utilizing offline datasets from a shifted linear model to accelerate online exploration in a target environment?*

In this paper, we provide an affirmative answer to this question by formulating a problem called hybrid linear bandits with biased offline data. We summarize our contributions as follows:

- **Algorithmic design.** We introduce a new ellipsoidal confidence set that incorporates biased offline data and propose the hybrid-LinUCB algorithm. Our regret upper bound takes the form $\tilde{O}(\min\{d\sqrt{T}, (\sqrt{\lambda_{\max}(V_0)}V + \sqrt{d})\sqrt{\frac{dT^2}{T+\lambda_{\min}(V_0)}}\})$, ensuring that the algorithm never performs worse than the purely-online baseline, while automatically leveraging even biased offline data whenever beneficial. Our analysis also unifies several regimes, covering online linear bandits, unbiased hybrid linear bandits, and biased hybrid multi-armed bandit (MAB) settings.

- **Refined bias–quality tradeoff.** Compared with prior work on hybrid biased MAB, our regret bound captures a more refined dependence on the offline data via the term $\sqrt{\frac{\lambda_{\max}(V_0)}{T+\lambda_{\min}(V_0)}}VT$, which directly quantifies the tradeoff between the offline data quality (size and structure) and the bias magnitude. Intuitively, a large $\lambda_{\max}(V_0)$ indicates strong coverage in certain directions, but also amplifies the effect of bias if it is concentrated there; in contrast, a large $\lambda_{\min}(V_0)$ reflects uniform coverage across directions, mitigating the impact of bias and enhancing robustness.

- **Analysis framework.** We provide a new analytical framework for hybrid linear bandits that refines matrix-norm based arguments in two complementary ways. First, instead of relying on aggregate quantities such as $\det(V_t)/\det(V_0)$ in LinUCB (Lattimore & Szepesvári, 2020), we show how the per-round action norm $\|a_t\|_{(V_t+V_0)^{-1}}$ can be decomposed across individual eigen-directions, thereby revealing how offline data contribute to exploration in each dimension. Second, we decompose the estimation error $\|\hat{\theta} - \theta^*\|_V$ into additive contributions from bias, online noise, offline noise, and regularization. These two perspectives offer a unified way to interpret the role of offline data and providing a bridge to link the bias $V$ and the quality of offline data $\lambda_i(V_0)$, yield richer results when specialized to hybrid MAB.

- **Lower bound.** We construct hard instances depending on the bias magnitude and establish matching lower bounds, showing that our upper bounds are near-optimal up to logarithmic factors.

- **Empirical validation.** We complement our theory with numerical simulations that compare hybrid-LinUCB against pure online and unbiased hybrid baseline. The results confirm our theoretical predictions: hybrid-LinUCB consistently benefits from offline data while remaining robust to bias.

Section 2 introduces the setting of hybrid linear bandits with biased offline data together with our assumptions. Section 3 presents the algorithmic design and underlying intuition, and Section 4 develops the regret bounds and compares them with prior work. Section 5 sketches the main ideas

of the proofs, and Section 6 concludes. Related work, detailed proofs, and experimental results are deferred to the Appendix.

## 2 PROBLEM SETUP

**Notation.** For any vector $a \in \mathbb{R}^d$ and any symmetric positive semidefinite matrix $A \in \mathbb{R}^{d \times d}$, we write $\|a\| := \sqrt{a^\top a}$, $\|a\|_A := \sqrt{a^\top A a}$. $\lambda_{\max}(A), \lambda_{\min}(A), \lambda_k(A)$ denote the largest, the smallest, and the $k$-th largest eigenvalue of matrix $A$, respectively.

**Online model and offline data.** We consider the *hybrid linear bandits* problem, which extends the classical linear bandits setting in online learning. In the standard formulation, at each round $t$, the learner selects an action $a_t \in \mathcal{A} \subseteq \mathbb{R}^d$ (where $\mathcal{A}$ may be finite or infinite) and observes a stochastic reward

$$x_t = \langle a_t, \theta_* \rangle + \eta_t, \qquad t = 1, 2, \ldots, T,$$

where $\theta_* \in \mathbb{R}^d$ is an unknown parameter and $\eta_t$ is zero-mean 1-subgaussian noise.

In the hybrid setting, the learner is granted access to an *offline dataset* prior to the online interaction. This dataset, denoted by $D_0$, consists of $N$ pairs $(b_s, y_s)$, where $b_s \in \mathcal{A}$ denotes an action and $y_s$ the corresponding feedback. Without loss of generality, we assume that the offline feedback also follows a linear structure, albeit potentially governed by a different parameter. Specifically, we assume

$$y_s = \langle b_s, \theta^{\text{off}} \rangle + \eta_s, \qquad s = 1, \ldots, N,$$

where $\theta^{\text{off}} \in \mathbb{R}^d$ is the offline parameter that may be different from $\theta_*$ and $\eta_s$ is zero-mean 1-subgaussian noise. Moreover, we assume that $\{\eta_t\}_{t=1}^T$ and $\{\eta_s\}_{s=1}^N$ are mutually independent and are independent of the learner's actions/history. The central challenge is to effectively integrate the biased offline data with online exploration to achieve improved regret guarantees. Overall, we define a instance $I = \{D_0, \theta^{\text{off}}, \theta_*, T\}$.

**Regret.** Let $a^* := \arg\max_{a \in \mathcal{A}} \langle a, \theta_* \rangle$ denote the optimal action under the true parameter. The cumulative regret after $T$ rounds is

$$\mathbb{E}[R_T] := \mathbb{E}\left[ \sum_{t=1}^T \left( \langle a^*, \theta_* \rangle - \langle a_t, \theta_* \rangle \right) \right].$$

**Bias upper bound.** The offline parameter may differ from the true parameter. We assume that the discrepancy is bounded in $\ell_2$ norm to characterizes the overall magnitude of bias in a global sense:

$$\|\theta_* - \theta^{\text{off}}\| \leq V, \text{ where } V \in R^+.$$

**Boundedness.** We assume $\|a\| \leq L$ for all $a \in \mathcal{A}$ (hence $\|b_s\| \leq L$) and $\|\theta_*\|, \|\theta^{\text{off}}\| \leq S$.

## 3 HYBRID LINUCB ALGORITHM

In this section, we present our proposed hybrid LinUCB algorithm (Algorithm 1) that adaptively balances the use of offline data to enhance the efficiency of online learning.

For convenience, define $V_t$ as the Gram matrix that collects purely online features and $V_{t,N}$ as the Gram matrix that collects action features from both online interaction and offline dataset. Similarly, denote $\hat{\theta}_t$ and $\hat{\theta}_{t,N}$ to represent the estimated parameter using pure online and the hybrid feedback (Line 7-8). We then construct two corresponding confidence sets $C_t^{\text{on}}$ and $C_t^{\text{hyb}}$ defined as follows:

$$\mathcal{C}_t^{\text{on}} = \{\theta : \|\theta - \hat{\theta}_t\|_{V_t} \leq \sqrt{\beta_t}\}, \quad \mathcal{C}_t^{\text{hyb}} = \{\theta : \|\theta - \hat{\theta}_{t,N}\|_{V_{t,N}} \leq \sqrt{\beta_{t,N}}\}.$$

The following lemma shows that by choosing an appropriate confidence radius, the real reward parameter will fall into the confidence sets with high probability.

---

**Algorithm 1** H-LinUCB with Intersected Confidence Sets

---

**Require:** action set $\mathcal{A} \subseteq \mathbb{R}^d$; offline data $D_0 = \{(b_s, y_s)\}_{s=1}^N$; regularization $\lambda > 0$; confidence $\delta \in (0, 1)$

1: $V_{1,N} \leftarrow \lambda I + \sum_{s=1}^N b_s b_s^\top, \hat{\theta}_{1,N} = V_{1,N}^{-1} \sum_{s=1}^N b_s y_s, V_1 = \lambda I, \hat{\theta}_1 = \mathbf{0}$
2: **for** $t = 1, 2, \ldots, T$ **do**
3:    compute $\sqrt{\beta_t}$ and $\sqrt{\beta_{t,N}}$ defined in equation 2
4:    **Action selection:**
5:    $a_t \leftarrow \arg\max_{a \in \mathcal{A}} \left\{ U_t^{\text{int}}(a) \right\}$ where

$$U_t^{\text{int}}(a) = \min\left\{ \underbrace{\langle a, \hat{\theta}_t \rangle + \sqrt{\beta_t} \|a\|_{V_t^{-1}}}_{\text{UCB from online ellipsoid}}, \underbrace{\langle a, \hat{\theta}_{t,N} \rangle + \sqrt{\beta_{t,N}} \|a\|_{V_{t,N}^{-1}}}_{\text{UCB from hybrid ellipsoid}} \right\}. \quad (1)$$

6:    Pull $a_t$ and observe $x_t$
7:    $V_{t+1} \leftarrow V_t + a_t a_t^\top, \quad \hat{\theta}_{t+1} \leftarrow V_{t+1}^{-1} \sum_{s=1}^t a_s x_s$
8:    $V_{t+1,N} \leftarrow V_{t,N} + a_t a_t^\top, \quad \hat{\theta}_{t+1,N} \leftarrow V_{t+1,N}^{-1} \left( \sum_{s=1}^N b_s y_s + \sum_{s=1}^t a_s x_s \right)$
9: **end for**

---

**Lemma 1.** *Choosing the confidence radius*

$$\sqrt{\beta_t} = \sqrt{\lambda} S + \sqrt{2\log(\tfrac{1}{\delta}) + d \cdot \log\left(\tfrac{d\lambda + tL^2}{d\lambda}\right)}, \sqrt{\beta_{t,N}} = \sqrt{\lambda_{\max}(V_0)} V + \sqrt{\beta_t} + \sqrt{\beta_N}. \quad (2)$$

*It holds that* $\Pr(\theta_* \in \mathcal{C}_t^{\text{on}} \cap \mathcal{C}_t^{\text{hyb}}) \geq 1 - \delta$.

The algorithm would then select the arm with the highest upper confidence bounds, i.e., $a_t \in \arg\max_{a \in \mathcal{A}} U_t^{\text{int}}(a)$ (Line 5), where $U_t^{\text{int}}(a)$ is defined as the minimum of two UCB estimates: one based solely on online data and the other incorporating both online and offline data (equation 1).

**Why the hybrid confidence radius is constructed this way.**    Recall that the hybrid ellipsoid is

$$\mathcal{C}_t^{\text{hyb}} = \left\{ \theta : \|\theta - \hat{\theta}_{t,N}\|_{V_{t-1,N}} \leq \sqrt{\beta_{t,N}} \right\}, \qquad V_{t-1,N} = V_{t-1} + V_0,$$

and Lemma 1 shows (with high probability)

$$\|\hat{\theta}_{t,N} - \theta^*\|_{V_{t-1,N}} \leq \underbrace{\sqrt{\lambda_{\max}(V_0)} \|\theta^* - \theta_{\text{off}}\|}_{\text{bias}} + \underbrace{\left\| \sum_{s=1}^{t-1} a_s \eta_s \right\|_{V_{t-1}^{-1}}}_{\text{online noise}} + \underbrace{\left\| \sum_{s=1}^N b_s \eta_s' \right\|_{V_0^{-1}}}_{\text{offline noise}} + \underbrace{\sqrt{\lambda} \|\theta^*\|}_{\text{ridge}}.$$

That is, the radius naturally decomposes into *bias*, *online noise*, *offline noise* and the *ridge* term (more computational details are provided in Appendix A.1). This decomposition is conceptually unavoidable: if we decide to exploit offline data in estimation, we must (i) pay an extra variance term arising from the offline responses and (ii) protect against the possible shift $\theta^* \neq \theta_{\text{off}}$ by inflating the radius with a bias term.

**"Is the hybrid radius always larger than the online one?"** Indeed, compared to the pure-online radius $\sqrt{\beta_t}$, the hybrid radius contains two additional addends (bias and offline noise). This, however, does *not* mean the hybrid UCB is always looser. The arm-level uncertainty that drives decisions is the product

$$\underbrace{\sqrt{\beta_{t,N}}}_{\text{radius inflation}} \times \underbrace{\|a\|_{V_{t-1,N}^{-1}}}_{\text{geometry (variance) shrinkage}} \qquad \text{vs} \quad \sqrt{\beta_t} \|a\|_{V_{t-1}^{-1}}.$$

While $\sqrt{\beta_{t,N}} \geq \sqrt{\beta_t}$, the geometry is strictly more favorable: since $V_{t-1,N} \succeq V_{t-1}$,

$$\|a\|_{V_{t-1,N}^{-1}} \leq \|a\|_{V_{t-1}^{-1}} \qquad (\forall a \in \mathbb{R}^d).$$

More precisely, letting $C_t = V_{t-1}^{-1/2} V_0 V_{t-1}^{-1/2}$ with eigenvalues $\{\lambda_{k,t}\}_{k=1}^d$ and writing $a$ in the corresponding basis gives the spectral identity

$$a^\top (V_{t-1} + V_0)^{-1} a = \sum_{k=1}^d \frac{\gamma_{k,t}^2}{1 + \lambda_{k,t}}, \qquad \gamma_{k,t} = (U_t^\top V_{t-1}^{-1/2} a)_k,$$

so each direction $k$ is shrunk by a factor $\frac{1}{1+\lambda_{k,t}}$. Hence offline coverage (large $\lambda_{k,t}$) *reduces* the prediction variance exactly along the directions where $V_0$ is strong. This shrinkage can more than offset the extra terms in $\sqrt{\beta_{t,N}}$.

**Centering effect (why $\hat{\theta}_{t,N}$ helps).** Besides width, the UCB also depends on the center. The hybrid estimator $\hat{\theta}_{t,N} = V_{t-1,N}^{-1} \left( \sum_{s=1}^N b_s y_s + \sum_{s=1}^{t-1} a_s x_s \right)$ aggregates *more information* than the pure-online one. When the bias level $\|\theta_* - \theta_{\text{off}}\| \le V$ is small, Lemma 1 yields

$$\|\hat{\theta}_{t,N} - \theta_*\|_{V_{t-1,N}} \le \sqrt{\lambda_{\max}(V_0)}\, V + \sqrt{\beta_t} + \sqrt{\beta_N},$$

and the first term vanishes when $V = 0$. In that unbiased (or small-bias) regime the center $\hat{\theta}_{t,N}$ is *closer to $\theta^*$ in the $V_{t-1,N}$-norm* because of the added offline evidence, while $\|a\|_{V_{t-1,N}^{-1}}$ is simultaneously smaller. Note that $\sqrt{\beta_t}$ grows only logarithmically with $t$ (standard self-normalized bounds), so—as in pure online LinUCB—the dominant effect with time is that the design matrix accumulates information and the estimator concentrates towards $\theta_*$.

**Summary.** The hybrid radius explicitly captures the trade-off between offline coverage (sample size and directional structure encoded by $V_0$) and bias $V$. When coverage is strong and bias small, the geometric shrinkage $\|a\|_{V_{t-1,N}^{-1}}$ and improved centering of $\hat{\theta}_{t,N}$ tighten the UCB, whereas with large bias the *min-of-two* rule safely reverts to the pure-online ellipsoid.

More specifically:

- If $V$ is *small* and/or the offline design is *well-conditioned* (large $\lambda_{\min}(V_0)$ in aligned directions), then the shrinkage $\|a\|_{V_{t-1,N}^{-1}} \ll \|a\|_{V_{t-1}^{-1}}$ and the improved centering of $\hat{\theta}_{t,N}$ outweigh the small increase in radius; the hybrid UCB becomes *tighter* and accelerates exploration.
- If $V$ is *large* and $V_0$ is also large, the inflation $\sqrt{\lambda_{\max}(V_0)}\, V$ indicates offline data may be misleading. In this case, the algorithm *automatically falls back* to the online UCB due to the intersected rule $U_t^{\text{int}}(a) = \min\left\{ \langle a, \hat{\theta}_t \rangle + \sqrt{\beta_t}\|a\|_{V_{t-1}^{-1}},\ \langle a, \hat{\theta}_{t,N} \rangle + \sqrt{\beta_{t,N}}\|a\|_{V_{t-1,N}^{-1}} \right\}$.

Overall, although the hybrid radius includes extra bias and offline-noise terms and can be larger than the pure-online one, the accompanying variance shrinkage and improved centering—both induced by $V_{t-1,N}$—often make the hybrid UCB *tighter* whenever the offline structure is useful, while the intersected design guarantees robustness against misleading offline data.

## 4 REGRET ANALYSIS

This section provides regret guarantees for hybrid linear bandits with potential bias as well as the corresponding discussions. Theorem 1 first provides a regret upper bound of Algorithm 1.

**Theorem 1.** *Following Algorithm 1 with a valid bias bound $V$, the regret satisfies*

$$\mathbb{E}[Reg(T)] \le \tilde{O}\left( \min\left\{ d\sqrt{T}, (\sqrt{\lambda_{\max}(V_0)}V + \sqrt{d}) \sqrt{\frac{dT^2 L^2}{TL^2 + \lambda_{\min}(V_0)}} \right\} \right), \qquad (3)$$

In the following, we discuss the main intuition and compare our results with related works.

**Intuition of the regret.** The upper bound highlights two key aspects. First, it contains the term $\sqrt{\lambda_{\max}(V_0)}V$, which indicates that if the offline dataset is large but highly biased, it may introduce

additional risk and inflate the regret. This shows that our algorithm is inherently "safe": once the bias $V$ is large, the algorithm automatically falls back to the pure-online route. Second, the result makes explicit the core trade-off: $\lambda_{\min}(V_0)$ quantifies the *coverage* of offline data in different directions, while $V$ captures its *misalignment*. Our regret bound precisely balances these two factors: large coverage with small bias yields significant improvements; large coverage with large bias prevents reliance on offline data; and small coverage limits the overall impact of offline data.

**Results in the reduced purely online setting or hybrid setting without bias.** Intuitively, the first term in equation 3 corresponds to the standard regret in the purely online setting, while the second term reflects the additional benefit obtained by leveraging hybrid feedback. When the offline dataset is empty, i.e., $\lambda_{\max}(V_0) = \lambda_{\min}(V_0) = 0$, the second term coincides with the first, and our result recovers the regret bound of the classic LinUCB algorithm. Conversely, when the offline dataset is fully aligned with the online setting, i.e., $V = 0$, the regret reduces to $\tilde{O}(d\sqrt{T^2L^2/(TL^2 + \lambda_{\min}(V_0))})$. This bound is strictly better than that of standard LinUCB whenever $\lambda_{\min}(V_0) > 0$.

*Remark 1. Our current analysis highlights the role of $\lambda_{\min}(V_0)$, which suggests that benefits arise whenever the offline design is full-rank. However, even when $V_0$ is not full rank, offline data can still be beneficial by providing coverage in certain directions, thereby accelerating exploration along those subspaces. Capturing such partial coverage benefits requires a more refined, dimension-wise characterization of the interaction between $V_0$ and the online trajectory. Similar ideas have been discussed in the context of bandits with partially observable confounded data (Tennenholtz et al., 2021), where one leverages coverage restricted to a subset of directions. We think similar technique and intuition can perform in hybrid linear bandits as well and leave a formal treatment of this setting as an interesting direction for future work.*

**Comparison with Zhang et al. (2025).** This work is the most closely related one that also investigates the hybrid linear bandit problem under potential bias. However, both its technical approach and its final results differ significantly from ours. Specifically, this work also constructs two confidence sets: one, as in our setting, based on the purely online dataset; the other, however, is not derived from the combined offline and online data but instead defined as an $\ell_2$-ball centered at $\theta^{\text{off}}$ with radius $V$. This construction does not effectively utilize the offline data, which is reflected in the regret bound by the absence of any improvement as the amount of offline data increases. Specifically, their regret order is $\tilde{O}(\min\{d\sqrt{T}, VT + \sqrt{d}T/\sqrt{\lambda_{\min}(V_0)}\})$. When no offline data are available, the second term (intended to capture the effect of offline information) fails to recover the purely online result. When offline data are available and $V = 0$, their result requires $\lambda_{\min}(V_0) > T$ in order to obtain any benefit. This contradicts the natural intuition that unbiased offline data should immediately enhance online learning performance. In contrast, our algorithm achieves such a benefit as soon as $\lambda_{\min}(V_0) > 0$, which is fully consistent with this intuition. Moreover, when $V \neq 0$, their bound suffers from a direct linear dependence on $VT$, while our result eliminates this dependence by more finely characterizing the influence of bias $V$ in terms of the quantity of offline data. Consequently, even when $V$ is large, our regret remains sub-linear in $T$ if the amount of offline data $\lambda_{\max}(V_0)$ is small, avoiding the overly pessimistic behavior exhibited in their bound.

**Comparison with Vijayan et al. (2025).** Vijayan et al. (2025) study hybrid linear bandits in the *unbiased* setting ($V = 0$) and design an elimination-based algorithm. Our approach targets a broader regime. First, we allow an *infinite* action set and do not require each offline sample to be collected independently, a fixed data-generating policy, or per-arm coverage conditions such as ensuring every arm in the support of $\pi_{\text{off}}$ receives at least $\widetilde{\Omega}(\log((T + T_{\text{off}})/d))$ samples. More importantly, we analyze the *biased* hybrid setting, which is both more general and technically more challenging.

In the unbiased case, the two approaches achieve regret bounds of the same order. However, Vijayan et al. (2025) obtain a finer decomposition that tracks the per-direction exploration through the eigenvalues $\{\lambda_i(V_0)\}$, whereas our current analysis aggregates offline coverage through $\lambda_{\min}(V_0)$. From this perspective, their bound is more granular.

Methodologically, elimination in linear bandits typically hinges on (re-)designing a G-optimal allocation across phases: given a living action set, one selects a well-spread subset of actions to control the worst-case prediction variance. After incorporating offline data, the design naturally adapts,

but G-optimality optimizes only the *geometry* of actions and does not directly encode the *mismatch* between offline and online data; consequently, extending elimination cleanly from the unbiased to the biased case is nontrivial and, to our knowledge, remains open. By contrast, our hybrid Lin-UCB constructs intersected confidence ellipsoids that explicitly capture the trade-off between bias and the size/structure of the offline data. This yields a transparent, interpretable mechanism and, importantly, imposes no finiteness restriction on the action set—making the approach amenable to downstream RL formulations.

*Remark 2. Vijayan et al. (2025) also discuss the connection to the warm-start analysis of LinUCB. By substituting $V_0 = T_{\text{off}} V_{\pi_{\text{off}}}$ into the framework of Valko et al. (2014) , one can recover regret bounds for LinUCB in the unbiased case. If we reduce our setting to the unbiased case, their formulation again provides richer conclusions, as it quantifies the exploration reduction along each eigen-direction of $V_0$ rather than aggregating via $\lambda_{\min}(V_0)$. This perspective might also inform some of the concerns and open problems we highlighted in the future work section.*

*Remark 3. Moreover, we emphasize that the supposed advantage of elimination over LinUCB in terms of $d$ versus $\sqrt{d}$ dependence does not hold in general. As rigorously discussed in Chapter 24 of the Lattimore & Szepesvári (2020), the $\sqrt{d}$ rate is tied to the assumption of an infinite (or continuous) action set. When the action set $\mathcal{A}$ is finite, say $|\mathcal{A}| = k$, the regret of LinUCB reduces from $\mathcal{O}(d\sqrt{T})$ to $\mathcal{O}(\sqrt{dT})$, eliminating the apparent gap.*

**Comparison with Cheung & Lyu (2024).** When our upper bound is reduced to the MAB setting, the key difference from the bound in Cheung & Lyu (2024) lies in the bias-related term. Their bound involves a $VT$ term, whereas ours takes the form $\tilde{O}(\sqrt{\frac{N_{\max}}{T+N_{\min}}} VT)$ which is strictly tighter whenever the maximum gap among offline arm sample counts does not exceed $T$. In the opposite regime, when the sample imbalance across arms is very large, our term can indeed be larger than theirs; yet it remains controlled by the magnitude of $V$, and never worse than the purely-online $\sqrt{T}$ dependence. Thus, the more important message is that our bound makes the trade-off between $N$ (offline sample allocation) and $V$ (bias) transparent.

In the linear bandit setting, this intuition becomes even more pronounced. A large $\lambda_{\max}(V_0)$ indicates that some directions are already well-explored by the offline data, but this also amplifies the effect of bias if it happens to concentrate along those directions. Meanwhile, $\lambda_{\min}(V_0)$ provides a coverage guarantee across all directions; increasing it ensures more uniform exploration and consequently diminishes the adverse effect of bias as the geometry of the action space becomes better understood. In this sense, our formulation highlights how the offline covariance structure governs the interplay between bias and online learning accuracy.

**Theorem 2.** *Let $\theta_* - \theta^{off} = (v_1, ..., v_d)^\top \in \mathbb{R}^d$, then $v_{\max} \le V/\sqrt{d}$, $\mathcal{A} = \{1/\sqrt{d}, -1/\sqrt{d}\}^d$ and $V_0$ be fixed but arbitrary. For any non-anticipatory policy $\pi$, there exists a Gaussian instance $\theta_*$ with offline design matrix $V_0$, such that*

$$\mathbb{E}[\text{Reg}(T)] \ge \min \left\{ d\sqrt{T}, \frac{e^{-\frac{S^2}{2d}}}{4} T \sum_{i=1}^d \frac{v_i}{\sqrt{T + \lambda_i(V_0)}} + \frac{e^{-\frac{1}{2}}}{4} T \sum_{i=1}^d \frac{1}{\sqrt{T + \lambda_i(V_0)}} \right\}. \quad (4)$$

**Intuition of lower bound.** The lower bound in Theorem 2 shows that, after uniformly controlling the $v_i$, the order matches our upper bound (up to logarithmic factors), establishing the near-optimality of our result. Importantly, the bound decomposes the effect of bias along each eigen-direction of $V_0$, thereby quantifying how different coordinates of the bias vector $v$ contribute to the online regret. In the unbiased case ($V = 0$), the lower bound continues to match the upper bound and, in fact, yields a slightly finer characterization. However, compared to Vijayan et al. (2025), this does not represent a substantive improvement: our bound can essentially be viewed as a special case of theirs, corresponding to an *average allocation* of offline samples across dimensions, whereas their result depends on the full distribution of offline data across coordinates and selects the optimal allocation. The comparison with other related works is similar with the upper bound part above.

# 5 PROOF SKETCH OF THEOREM

## 5.1 PROOF SKETCH OF THEOREM 1

We outline the key steps of the regret analysis when the algorithm selects actions based on the hybrid confidence ellipsoid. Detailed proof are provided in Appendix C.

Step 1: Regret decomposition. At each round $t$, the instantaneous regret satisfies

$$r_t = \langle a^*, \theta^* \rangle - \langle a_t, \theta^* \rangle \le \min \left\{ \|a_t\|_{V_{t-1}^{-1}} \|\hat{\theta}_t - \theta^*\|_{V_{t-1}}, \ \|a_t\|_{V_{t-1,N}^{-1}} \|\hat{\theta}_{t,N} - \theta^*\|_{V_{t-1,N}} \right\}.$$

Step 2: Confidence sets. By Lemma 1, with high probability we have

$$\|\hat{\theta}_t - \theta^*\|_{V_{t-1}} \le \sqrt{\beta_t}, \qquad \|\hat{\theta}_{t,N} - \theta^*\|_{V_{t-1,N}} \le \sqrt{\beta_{t,N}}.$$

*Idea of Lemma 1.* Start from $\hat{\theta}_{t,N} - \theta^* = V_{t,N}^{-1} \left( \sum_{s=1}^{t} a_s x_s + \sum_{s=1}^{N} b_s y_s \right) - \theta^*$, and expand $x_s = \langle a_s, \theta^* \rangle + \eta_s, y_s = \langle b_s, \theta_{\text{off}} \rangle + \eta_s'$ to obtain

$$\hat{\theta}_{t,N} - \theta^* = V_{t,N}^{-1} \left[ V_0(\theta_{\text{off}} - \theta^*) + \sum_{s=1}^{t} a_s \eta_s + \sum_{s=1}^{N} b_s \eta_s' - \lambda \theta^* \right].$$

Using $\|B^{-1}c\|_B = \|c\|_{B^{-1}}$ and triangle inequality gives

$$\|\hat{\theta}_{t,N} - \theta^*\|_{V_{t-1,N}} \le \underbrace{\|V_0(\theta_{\text{off}} - \theta^*)\|_{V_{t-1,N}^{-1}}}_{\text{bias}} + \underbrace{\left\| \sum_{s=1}^{t} a_s \eta_s \right\|_{V_{t-1}^{-1}}}_{\text{online noise}} + \underbrace{\left\| \sum_{s=1}^{N} b_s \eta_s' \right\|_{V_0^{-1}}}_{\text{offline noise}} + \underbrace{\sqrt{\lambda} \|\theta^*\|}_{\text{ridge}}.$$

For the bias term, $V_{t-1,N} \succeq V_0$ implies $\|V_0(\theta_{\text{off}} - \theta^*)\|_{V_{t-1,N}^{-1}} \le \sqrt{\lambda_{\max}(V_0)} \|\theta_{\text{off}} - \theta^*\| \le \sqrt{\lambda_{\max}(V_0)} V$. For the two noise terms, apply self-normalized bounds (LinUCB/OFUL standard) to get

$$\left\| \sum_{s=1}^{t} a_s \eta_s \right\|_{V_{t-1}^{-1}} \le \sqrt{2 \log(1/\delta)} + d \log \left( 1 + \tfrac{tL^2}{d\lambda} \right), \ \left\| \sum_{s=1}^{N} b_s \eta_s' \right\|_{V_0^{-1}} \le \sqrt{2 \log(1/\delta)} + d \log \left( 1 + \tfrac{NL^2}{d\lambda} \right).$$

Combining the four pieces and using $\|\theta^*\| \le S$ yields $\|\hat{\theta}_{t,N} - \theta^*\|_{V_{t-1,N}} \le \sqrt{\lambda_{\max}(V_0)} V + \sqrt{\beta_t} + \sqrt{\beta_N}$, i.e., the stated hybrid radius.

Step 3: Controlling the hybrid norm. The key quantity is $\|a_t\|_{V_{t-1,N}^{-1}}^2 = a_t^\top (V_{t-1} + V_0)^{-1} a_t$. Using the matrix inversion identity $A^{-1} - (A + B)^{-1} = A^{-1}B(A + B)^{-1}$ and eigen-decomposition, we obtain

$$a_t^\top (V_{t-1} + V_0)^{-1} a_t = \sum_{k=1}^{d} \frac{1}{1 + \lambda_{k,t}} \gamma_{k,t}^2,$$

where $\lambda_{k,t}$ are eigenvalues of $V_{t-1}^{-1/2} V_0 V_{t-1}^{-1/2}$ and $\gamma_{k,t}$ are corresponding coordinates.

Step 4: Eigenvalue lower bound. Lemma 2 ensures $\lambda_{k,t} \ge \frac{\lambda_k(V_0)}{\lambda_{\max}(V_{t-1})} \ge \frac{\lambda_k(V_0)}{tL^2}$. Hence $\frac{1}{1 + \lambda_{k,t}} \le \frac{1}{1 + \lambda_k(V_0)/(tL^2)}$, showing that offline data regularizes the effective variance.

Step 5: Summation. The standard bound $\sum_{t=1}^{T} a_t^\top V_{t-1}^{-1} a_t = \tilde{O}(d)$ implies

$$\sum_{t=1}^{T} a_t^\top (V_{t-1} + V_0)^{-1} a_t \le \sum_{t,k} \frac{\gamma_{k,t}^2}{1 + \lambda_k(V_0)/(tL^2)}.$$

Step 6: Aggregation and final bound. Applying Cauchy–Schwarz yields and Substituting the above estimates, and using the regret analysis for pure online term gives

$$\sum_{t=1}^{T} r_t \le \sqrt{T \cdot \beta_{T,N} \cdot \sum_{t=1}^{T} a_t^\top (V_{t-1} + V_0)^{-1} a_t}.$$

Step 7: Final bound. Substituting the above estimates, and using the regret analysis for pure online term gives

$$\text{Reg}(T) = \tilde{O}\left(\min\left\{d\sqrt{T}, \ \left(\sqrt{\lambda_{\max}(V_0)}V + \sqrt{d}\right)\sqrt{\tfrac{dT^2L^2}{TL^2 + \lambda_{\min}(V_0)}}\right\}\right).$$

## 5.2 PROOF SKETCH OF THEOREM 2

At a high level, the lower bound is obtained by reducing the problem to multiple hypothesis testing. Biased offline data can mislead the learner along certain directions, while in orthogonal subspaces the problem reduces to the purely online case. We construct hard instances differing only in selected coordinate signs, so that each misclassification incurs fixed regret and information-theoretic bounds limit their frequency. This reveals the trade-off: offline data accelerates learning when aligned with the true parameter, but may amplify regret if the bias lies in high-coverage directions.

Step 1: Hard instance construction. We consider parameters $\theta(u) = \epsilon u$ with $u \in \{\pm 1\}^d$ and symmetric action set $A = \{\pm \frac{1}{\sqrt{d}}\}^d$. The offline data are generated from $\theta_{\text{off}}$, while the online rewards follow $\theta^*$, with $\|\theta^* - \theta_{\text{off}}\| \leq V$.

Step 2: Misclassification–regret link. The regret can be expressed as a weighted count of sign misclassifications across coordinates, so bounding the expected number of mistakes directly yields a regret lower bound:

$$\text{Reg}_t = \langle \theta_{u^{\text{on}}}, a^*(u^{\text{on}}) - a_t \rangle = \frac{2}{\sqrt{d}} \sum_{i=1}^{d} \epsilon_i I\{\text{sgn}(a_t) \neq u_i^{\text{on}}\}$$

Step 3: Information-theoretic reduction. For each coordinate $i$, we construct neighboring instances that differ only on the $i$-th coordinate of the online/offline parameters. Depending on the bias size, we choose $\epsilon_i$ so that the KL divergence between the two instances is small, forcing a constant misclassification probability by summing up the difference from online distributions and offline distributions.

Step 4: Bretagnolle–Huber inequality and aggregation. Applying Bretagnolle–Huber converts the KL bound into a uniform lower bound on the expected number of mistakes. And summing over $d$ coordinates gives

$$\mathbb{E}[\text{Reg}(T)] \ \geq \ \Omega\left(\min\left\{d\sqrt{T}, \ T\sum_{i=1}^{d} \frac{v_i}{\sqrt{T + \lambda_i(V_0)}} + T\sum_{i=1}^{d} \frac{1}{\sqrt{T + \lambda_i(V_0)}}\right\}\right).$$

# 6 CONCLUSION

We studied the problem of hybrid linear bandits with biased offline data, and proposed the hybrid-LinUCB algorithm based on intersected confidence sets. Our analysis shows that the algorithm automatically interpolates between purely online learning and unbiased hybrid learning, while gracefully handling biased offline datasets. We established both upper and lower bounds, demonstrating near-optimality and highlighting the tradeoff between offline data quality and bias. Our framework not only generalizes existing results for online, unbiased, and MAB settings, but also provides new analytical tools for decomposing estimation error and regret in hybrid bandits.

**Future work.** Several open directions remain:

- **Dimension-wise characterization.** Our current bounds rely on $\lambda_{\min}(V_0)$; a finer analysis using all eigenvalues $\{\lambda_i(V_0)\}$ could capture partial coverage benefits of offline data.
- **Vector-valued bias.** We model the bias $V$ via its $\ell_2$ norm, a global measure. Considering $V$ as a vector could reveal richer, coordinate-dependent behaviors.
- **Beyond linear bandits.** Since linear models serve as a key building block for general function approximation, it is natural to ask whether our techniques can extend to more general function classes.

## ETHICS STATEMENT

No ethical concerns.

## REPRODUCIBILITY STATEMENT

Our experiments are based on small-scale synthetic simulations rather than large real-world datasets. We have therefore specified all details (including offline data generation, action set construction, and evaluation protocol) in the paper, such that the results can be reproduced directly without requiring code release.

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

# A RELATED WORK

**Online and Offline Linear Bandits.** The linear bandits problem is a generalization of the multi-armed bandit (MAB) setting, and has been widely studied due to its capacity to model many real-world applications. The online linear bandit problem dates back to Dani et al. (2008), who extend Upper Confidence Bound (UCB) algorithms to the stochastic linear bandit setting. Later, Abbasi-Yadkori et al. (2011) develop the OFUL algorithm, achieving a regret upper bound of $\tilde{\mathcal{O}}(d\sqrt{T})$ for purely online regret minimization (without any offline data). Beyond UCB-type methods, Agrawal & Goyal (2013) employ Thompson Sampling in linear contextual bandits. Moreover, to address drawbacks of sub-optimal bounds in settings with sub-exponential arm sets, a phased elimination algorithm has been proposed (see, e.g., Lattimore & Szepesvári (2020)). In contrast, the offline linear bandits problem assumes that the agent has access to an offline dataset and aims to identify the optimal action using that data. For example, Rashidinejad et al. (2021) propose a pessimism-based algorithm for offline linear bandits and Markov decision processes. Similarly, Li et al. (2022) propose pessimistic learning rules for offline linear contextual bandits and prove the sub-optimality gap for fixed contexts.

**Hybrid Bandits.** The hybrid bandits problem is closely related to the warm-start bandits setting, which studies how to use offline data to accelerate online exploration. Several works address this in the MAB setting. Shivaswamy & Joachims (2012) develop the HUCB algorithm, which exploits offline (historical) data to better approximate the confidence bounds in UCB-style schemes. Hao et al. (2023) propose a warm-started Thompson Sampling approach that attains a Bayesian regret bound. However, these works assume that the offline data and the online samples follow the same reward distribution—a strong assumption that often fails in practice. To relax this, Cheung & Lyu (2024) study a hybrid MAB problem in which the offline dataset may follow a different reward distribution than the online environment, and propose a MIN-UCB method that achieves a near-optimal regret bound given prior knowledge of the distribution shift.

As for the hybrid linear bandits problem, most of existing works still focus on the unbiased reward distribution setting. Oetomo et al. (2023) extend ideas from hybrid MAB to linear contextual bandits by developing both Thompson Sampling– and LinUCB–based hybrid algorithms. Vijayan et al. (2025) introduce an Offline-Online Phased Elimination (OOPE) algorithm for stochastic linear bandits. We notice this work has restrictive assumptions on the independence of offline data and finite arm set. There are only few works focusing on hybrid linear bandits with biased reward distribution. Zhang et al. (2025) study a biased parametric model for dynamic pricing (a special case of linear bandits) and also discuss the broader linear bandit setting. Their rate scales as $\tilde{O}(\sqrt{d}T/\sqrt{\lambda_{\min}(V_0)})$ when $V = 0$, which only improves upon the standard $O(d\sqrt{T})$ bound when $\lambda_{\min}(V_0)$ is sufficiently large; without such coverage, no uniform improvement is ensured.

**Hybrid Reinforcement Learning.** The task of utilizing offline dataset to enhance the performance of online exploration has also been investigated broadly in the realm of Reinforcement Learning (RL) with the name of Hybrid RL. Most of existing works consider this hybrid RL problem under the general function approximation setting. For instance, Song et al. (2023) propose hybrid Q-learning, and Tan & Xu (2024) develop a Global Optimism based on Local Fitting (GOLF)-based algorithm with general Q-function approximation. We point out that the general function approximation setting can recover the linear MDP setting (Wagenmaker & Pacchiano, 2023; Tan et al., 2024). However, an assumption commonly used in these works is that the offline and online MDPs are the same—that is, they share the same transitions and reward functions—which may be overly restrictive for many real-world RL applications. A recent work (Qu et al., 2025) addresses distribution shift in offline and online MDPs, terming this hybrid transfer RL. The authors propose the HySRL method to tackle this challenge. Nevertheless, this work only considers the tabular MDP without extending the proposed method to the general function approximation setting, and assume a known distribution shift lower bound between the offline and online environments.

# B IMPORTANT LEMMAS

**Lemma 2.** *For any positive matrix A and B, we have:*

$$\lambda_k(A^{-1/2}BA^{-1/2}) \geq \frac{\lambda_k(B)}{\lambda_{\max}(A)}$$

## B.1 PROOF OF LEMMA 1

*Proof.* Notice that:

$$\hat{\theta}_{t,N} - \theta_* = V_{t,N}^{-1}\left(\sum_{s=1}^{t} a_s x_s + \sum_{s=1}^{N} b_s y_s\right) - \theta_*$$

$$= V_{t,N}^{-1}\sum_{s=1}^{t} a_s\left(\theta_*^\top a_s + \eta_s\right) + V_{t,N}^{-1}\sum_{s=1}^{N} b_s\left(\theta_{\mathrm{off}}^\top b_s + \eta_s'\right) - \theta_*$$

$$\leq V_{t,N}^{-1}\left[-V_{t,N}\theta_* + \sum_{s=1}^{t} a_s a_s^\top \theta_* + \sum_{s=1}^{N} b_s b_s^\top \theta_* + \sum_{s=1}^{t} a_s \eta_s + \sum_{s=1}^{N} b_s \eta_s' + \sum_{s=1}^{N} b_s b_s^\top(\theta_{\mathrm{off}} - \theta_*)\right]$$

$$= V_{t,N}^{-1}\left[\sum_{s=1}^{N} b_s b_s^\top(\theta_{\mathrm{off}} - \theta_*) + \sum_{s=1}^{t} a_s \eta_s + \sum_{s=1}^{N} b_s \eta_s' - \lambda\theta_*\right].$$

Let $a, b \in \mathbb{R}^d$ be arbitrary vectors and $B \in \mathbb{R}^{d\times d}$ be symmetric positive definite (SPD). If $a = B^{-1}b$, then

$$\|a\|_B^2 := a^\top Ba = (B^{-1}b)^\top B(B^{-1}b) = b^\top B^{-1}b,$$
$$\Rightarrow \quad \|a\|_B = \sqrt{b^\top B^{-1}b} = \|b\|_{B^{-1}}.$$

Then we have:

$$\|\theta_* - \hat{\theta}_{t,N}\|_{V_{t,N}} = \left\|V_0(\theta_{\mathrm{off}} - \theta_*) + \sum_{s=1}^{t} a_s \eta_s + \sum_{s=1}^{N} b_s \eta_s' - \lambda\theta_*\right\|_{V_{t,N}^{-1}}$$

$$\leq \|V_0(\theta_{\mathrm{off}} - \theta_*)\|_{V_{t,N}^{-1}} + \left\|\sum_{s=1}^{t} a_s \eta_s\right\|_{V_{t,N}^{-1}} + \left\|\sum_{s=1}^{N} b_s \eta_s'\right\|_{V_{t,N}^{-1}} + \sqrt{\lambda}S$$

$$\leq \|V_0(\theta_{\mathrm{off}} - \theta_*)\|_{V_{t,N}^{-1}} + \left\|\sum_{s=1}^{t} a_s \eta_s\right\|_{V_t^{-1}} + \left\|\sum_{s=1}^{N} b_s \eta_s'\right\|_{V_0^{-1}} + \sqrt{\lambda}S$$

$$\leq \|V_0(\theta_{\mathrm{off}} - \theta_*)\|_{V_{t,N}^{-1}} + \sqrt{2\log\left(\tfrac{1}{\delta}\right) + d\log\left(1 + \tfrac{tL^2}{d\lambda}\right)}$$

$$+ \sqrt{2\log\left(\tfrac{1}{\delta}\right) + d\log\left(1 + \tfrac{NL^2}{d\lambda}\right)} + \sqrt{\lambda}S$$

$$\leq \sqrt{\lambda_{\max}(V_0)} \cdot V + \sqrt{\beta_t} + \sqrt{\beta_N},$$

where the fourth inequality comes from Theorem 20.4 of Lattimore & Szepesvári (2020) with probability of $1 - \delta$.

Along with pure online discussion on Theorem 20.5 of Lattimore & Szepesvári (2020), we have $\Pr(\theta_* \in \mathcal{C}_t^{\mathrm{on}} \cap \mathcal{C}_t^{\mathrm{hyb}}) \geq 1 - \delta$.

$\square$

### B.2 PROOF OF LEMMA 2

*Proof.* Let $U_k$ be a $k$-dimensional subspace spanned by the top-$k$ eigenvectors of $B$. For any $u \in U_k$ with $u \neq 0$, we have the Rayleigh bound

$$\frac{u^\top B u}{\|u\|^2} \geq \lambda_k(B).$$

Consider the $k$-dimensional subspace $S := A^{1/2} U_k$ and any $x \in S$ of the form $x = A^{1/2} u$ with $u \in U_k$, $x \neq 0$. Then

$$x^\top A^{-1/2} B A^{-1/2} x = u^\top B u \geq \lambda_k(B) \|u\|^2.$$

Moreover,

$$\|x\|^2 = u^\top A u \leq \lambda_{\max}(A) \|u\|^2 \quad \Rightarrow \quad \|u\|^2 \geq \frac{\|x\|^2}{\lambda_{\max}(A)}.$$

Hence for every nonzero $x \in S$,

$$\frac{x^\top A^{-1/2} B A^{-1/2} x}{\|x\|^2} \geq \frac{\lambda_k(B)}{\lambda_{\max}(A)}.$$

Taking the minimum over unit vectors in $S$ and then the maximum over all $k$-dimensional subspaces via the Courant–Fischer theorem yields

$$\lambda_k\big(A^{-1/2} B A^{-1/2}\big) = \max_{\dim S = k} \min_{\substack{x \in S \\ \|x\|=1}} x^\top A^{-1/2} B A^{-1/2} x \geq \frac{\lambda_k(B)}{\lambda_{\max}(A)}.$$

$\square$

## C PROOF OF THEOREM 1

### C.1 UNBIASED CASE

*Proof.* To make our proof easier to understand and read, we firstly considering the unbiased case. When $V = 0$, suppose that $U_t^{\mathrm{int}}(a_t) \leq \langle a, \hat{\theta}_{t,N} \rangle + \sqrt{\beta_{t,N}} \|a\|_{V_{t,N}^{-1}} := \langle a_t, \tilde{\theta}_{t,N} \rangle$. Then we have:

$$\langle \theta_*, a_t \rangle \leq \mathrm{UCB}_t(a^*) \leq \mathrm{UCB}_t(a_t) = \langle \tilde{\theta}_t, a_t \rangle.$$

Using Cauchy–Schwarz inequality and the assumption that $\theta_* \in \mathcal{C}_{t,N}$ and facts that $\tilde{\theta}_t \in \mathcal{C}_{t,N}$ leads to

$$r_t = \langle \theta_*, a^* \rangle - \langle \tilde{\theta}_{t,N} - \theta_*, a_t \rangle \leq \|a_t\|_{V_{t-1,N}^{-1}} \|\tilde{\theta}_{t,N} - \theta_*\|_{V_{t-1,N}}, \tag{5}$$

$$\text{where} \quad V_{t-1,N} = \sum_{s=1}^{t-1} a_s a_s^\top + \sum_{s=1}^{N} b_s b_s^\top := V_{t-1} + V_0.$$

$\|\tilde{\theta}_{t,N} - \theta_*\|_{V_{t-1,N}}$ can be controlled by $\sqrt{\beta_{t,N}} = \sqrt{\beta_t} + \sqrt{\beta_N}$.

Further control $\|a_t\|_{V_{t-1,N}^{-1}}$:

We fix $t$, define

$$\Delta_t(a_t) := a_t^\top V_{t-1}^{-1} a_t - a_t^\top (V_{t-1} + V_0)^{-1} a_t$$
$$= a_t^\top V_{t-1}^{-1} V_0 (V_{t-1} + V_0)^{-1} a_t.$$

(Matrix inversion identity: $A^{-1} - (A + B)^{-1} = A^{-1}B(A + B)^{-1}$)

Let $C_t = V_{t-1}^{-1/2}V_0 V_{t-1}^{-1/2}$, and $y_t = V_{t-1}^{-1/2}a_t$. Then

$$\Delta_t(a_t) = y_t^\top C_t (I + C_t)^{-1} y_t.$$

Consider spectral decomposition: $C_t = U_t \Lambda_t U_t^\top$. Let $z_t = U_t^\top y_t$, then

$$\Delta_t(a_t) = z_t^\top \Lambda_t (I + \Lambda_t)^{-1} z_t.$$

Since $U_t$ is orthogonal, $\|y_t\| = \|z_t\|$. Write $z_t = (\gamma_{1,t}, \ldots, \gamma_{d,t})^\top$, then

$$\Delta_t(a_t) = \sum_{k=1}^d \frac{\lambda_{k,t}}{1 + \lambda_{k,t}} \gamma_{k,t}^2,$$

where $\lambda_{k,t}$ is the k-th largest eigenvalue of $C_t$.

And also $a_t^\top V_t^{-1} a_t = y_t^\top y_t = \|y_t\|^2 = \|z_t\|^2 = \sum_{k=1}^d \gamma_{k,t}^2$.

Then

$$a_t^\top (V_{t-1} + V_0)^{-1} a_t = a_t^\top V_t^{-1} a_t - \Delta_t(a_t)$$

$$= \sum_{k=1}^d \gamma_{k,t}^2 - \sum_{k=1}^d \frac{\lambda_{k,t}}{1 + \lambda_{k,t}} \gamma_{k,t}^2$$

$$= \sum_{k=1}^d \frac{1}{1 + \lambda_{k,t}} \gamma_{k,t}^2.$$

From Ch.19 lemma19.4 in bandits book (Lattimore & Szepesvári, 2020):

$$\sum_{t=1}^T \|a_t\|_{V_{t-1}}^2 = \sum_{t=1}^T a_t^\top V_{t-1}^{-1} a_t$$

$$\leq 2 \log\left(\frac{\det V_T}{\det V_0}\right)$$

$$\leq 2d \cdot \log\left(\frac{d\lambda + TL^2}{d \det(\lambda I)^{1/d}}\right)$$

$$= \tilde{O}(d).$$

Hence

$$\sum_{t=1}^T \sum_{k=1}^d \gamma_{k,t}^2 = \sum_{t=1}^T a_t^\top V_{t-1}^{-1} a_t = \tilde{O}(d).$$

As a result,

$$\sum_{t=1}^T a_t^\top (V_{t-1} + V_0)^{-1} a_t = \sum_{t=1}^T \sum_{k=1}^d \frac{1}{1 + \lambda_{k,t}(V_{t-1}^{-1/2}V_0 V_{t-1}^{-1/2})} \gamma_{k,t}^2.$$

From Lemma 2: for all $t = 1, \ldots, T$, $k = 1, \ldots, d$,

$$\lambda_{k,t}(V_{t-1}^{-1/2}V_0 V_{t-1}^{1/2}) \geq \frac{\lambda_k(V_0)}{\lambda_k(V_{t-1})} \geq \frac{\lambda_k(V_0)}{t \cdot L^2}.$$

Thus, for function $f(x) = \frac{1}{1+x}$ decreasing,

$$\sum_{t=1}^{T}\sum_{k=1}^{d} \frac{1}{1 + \lambda_{k,t}(V_{t-1}^{-1/2}V_0 V_{t-1}^{1/2})}\,\gamma_{k,t}^2 \leq \sum_{t=1}^{T}\sum_{k=1}^{d} \frac{1}{1 + \frac{\lambda_k(V_0)}{(t-1)L^2}}\,\gamma_{k,t}^2.$$

Assume that

$$f(k,t) = \frac{1}{1 + \frac{\lambda_k(V_0)}{tL^2}}, \qquad g(k,t) = \gamma_{k,t}^2,$$

with $\lambda_k(V_0) > 0$ for all $k$, and

$$\sum_{t=1}^{T}\sum_{k=1}^{d} g(k,t) \leq \tilde{O}(d).$$

Then we have:

$$
\begin{aligned}
\mathbb{E}[\text{Reg(T)}] = \sum_{t=1}^{T} r_t &\overset{(a)}{\leq} \sum_{t=1}^{T} \sqrt{a_t^\top V_{t-1,N}^{-1} a_t} \cdot \sqrt{\beta_{t,N}} \\
&\overset{(b)}{\leq} \sqrt{T \sum_{t=1}^{T} a_t^\top V_{t-1,N}^{-1} a_t} \cdot \sqrt{\beta_{T,N}} \\
&\overset{(c)}{=} \sqrt{T \cdot \beta_{T,N}} \cdot \sqrt{\sum_t \sum_k fg} \\
&\overset{(d)}{\leq} \sqrt{T \cdot \beta_{T,N}} \cdot \sqrt{f_{\max} \sum_t \sum_k g} \\
&\overset{(e)}{\leq} \sqrt{T \cdot \beta_{T,N}} \cdot \tilde{O}\left( \sqrt{\frac{TL^2}{TL^2 + \lambda_{\min}(V_0)} \cdot d} \right) \\
&= \tilde{O}\left( \sqrt{\beta_{T,N} \frac{dT^2 L^2}{TL^2 + \lambda_{\min}(V_0)}} \right),
\end{aligned}
$$

where (a) is obtained by (5), (b) is from Cauchy-Schwarz inequality over $t$, (c) is from definition of $f$ and $g$, (d) is from the operation that $f_{\max} = \max_t \max_k f(k,t)$ and (e) is from $\sum_t \sum_k g \leq \tilde{O}(d)$.

$\square$

## C.2 BIASED CASE

*Proof.* Now we only need to consider the pure online term and the case when $V \neq 0$:

$$
\begin{aligned}
\sum_t r_t &\leq \sum_t \min\left\{ \|a_t\|_{V_t^{-1}} \|\theta^* - \hat{\theta}_t\|_{V_t}, \ \|a_t\|_{V_{t,N}^{-1}} \|\theta^* - \hat{\theta}_{t,N}\|_{V_{t,N}} \right\} \\
&\leq \sum_t \min\left\{ \|a_t\|_{V_t^{-1}} \cdot \sqrt{\beta_t}, \ \|a_t\|_{V_{t,N}^{-1}} \cdot \sqrt{\beta_{t,N}} \right\} \\
&\leq \min\left\{ \tilde{O}\left( \sqrt{\beta_t \cdot d \cdot T} \right), \ \tilde{O}\left( \sqrt{\beta_{t,N} \cdot d \cdot \frac{TL^2}{TL^2 + \lambda_{\min}(V_0)}} \right) \right\} \\
&\leq \min\left\{ \tilde{O}\left( \sqrt{\beta_T dT} \right), \tilde{O}\left( (\sqrt{\lambda_{\max}(V_0)}V + \sqrt{\beta_T} + \sqrt{\beta_N})\sqrt{\frac{dT^2 L^2}{TL^2 + \lambda_{\min}(V_0)}} \right) \right\}
\end{aligned}
$$

$\square$

## D  PROOF OF THEOREM 2

*Proof.* **Histories and filtration:**

Let $D_0 = \{(b_s, y_s)\}_{s=1}^N$ denote the offline dataset, possibly random. Define the filtration $\mathbb{F} = (\mathcal{H}_t)_{t \geq 0}$ by

$$\mathcal{H}_0 = \sigma(D_0), \qquad \mathcal{H}_t = \sigma(D_0, a_1, x_1, \ldots, a_t, x_t) \quad (t \geq 1),$$

where at round $t$ the learner selects $a_t \in \mathcal{A} \subseteq \mathbb{R}^d$ according to a (possibly randomized) policy $\pi_t(\cdot \mid \mathcal{H}_{t-1})$, and then receives

$$x_t = \langle a_t, \theta_* \rangle + \eta_t,$$

with $(\eta_t)_{t \geq 1}$ being conditionally zero-mean sub-Gaussian noises (independent of the past given $\mathcal{H}_{t-1}$). By construction, $a_t$ is $\mathcal{H}_{t-1}$-measurable (under $\pi_t$), and $x_t$ is $\mathcal{H}_t$-measurable.

**Define action sequence and feedback sequence:**

$$a_{1:T} = (a_1, \ldots, a_T), \ x_{1:T} = (x_1, \ldots, x_T)$$

**Policy:**

$$\{\pi_t\}_{t=1}^T, \ a_t \sim \pi_t(\cdot \mid \mathcal{H}_{t-1}, D_0)$$

(depends on $\mathcal{H}_{t-1}$ and offline matrix $D_0$)

**$\theta$ family and action set:**

$$\theta(u) = \epsilon u = \sum_{i=1}^d \epsilon_i u_i e_i, \text{where } \epsilon > 0, u \in \{1, -1\}^d \text{ and } \|\theta\|_2 \leq S.$$

$$\mathcal{A} = \{a \in \{1/\sqrt{d}, -1/\sqrt{d}\}^d\}, \|a\| \leq L.$$

Later we will use $u_1^{\mathrm{on}}, u_1^{\mathrm{off}}, u_2^{\mathrm{on}}, u_2^{\mathrm{off}}$ to distinguish the two different distribution between online environment and offline data distribution in case 1 and 2 correspondingly.

We have:

$$a^*(u^{\mathrm{on}}) = \frac{\mathrm{sgn}(\theta_{u^{\mathrm{on}}})}{\sqrt{d}} = \frac{u^{\mathrm{on}}}{\sqrt{d}} \in \mathcal{A},$$

$$\mathrm{Reg}_t = \langle \theta_{u^{\mathrm{on}}}, a^*(u^{\mathrm{on}}) - a_t \rangle = \frac{2}{\sqrt{d}} \sum_{i=1}^d \epsilon_i I\{\mathrm{sgn}(a_t) \neq u_i^{\mathrm{on}}\}$$

**Define the misclassification count:** Since in hybrid learning, the event $I\{\mathrm{sgn}(a_t) \neq u_i\}$ happens or not usually depend on both $u^{\mathrm{on}}$ and $u^{\mathrm{off}}$ (We remark that our discussion in here assume that the algorithm will use offline data, the pure online lower bound can obtain by the similar analysis and we do not further discuss in here), we define:

$$N_i^-(u^{\mathrm{on}}, u^{\mathrm{off}}) = \sum_{t=1}^T \mathbb{I}\{\mathrm{sgn}(a_t) \neq u_i^{\mathrm{on}}\}$$

**Then for the cumulative regret:**

$$\mathrm{Reg}(T) = \sum_{t=1}^T r_t = \frac{2}{\sqrt{d}} \sum_{i=1}^d \epsilon_i \, \mathbb{E}_{u^{\mathrm{on}}, u^{\mathrm{off}}} \big[ N_i^-(u^{\mathrm{on}}, u^{\mathrm{off}}) \big]$$

**Conditional reward distribution:**

$$\text{offline data } P_{u^{\mathrm{off}}}(y_s \mid b_s) = \mathcal{N}(\langle \theta_{u^{\mathrm{off}}}, b_s \rangle, \sigma^2),$$

$$\text{online feedback } P_{u^{\mathrm{on}}}(x_t \mid a_t) = \mathcal{N}(\langle \theta_{u^{\mathrm{on}}}, a_t \rangle, \sigma^2)$$

**Offline dataset distribution:**

$$P_{u^{\mathrm{off}}}(D_0) = \prod_{s=1}^N \nu(b_s) \mathcal{N}(\langle \theta_u^{\mathrm{off}}, b_s \rangle, \sigma^2)$$

**Case 1**  When $\sum_i v_i \geq e^{-\frac{1}{2}(S^2/d - 1)}d$, then

$$\Omega\Big(\frac{e^{-S^2/2d}}{4}T\sum_{i=1}^{d}\frac{v_i}{\sqrt{T + \lambda_i(V_0)}} + \frac{e^{-1/2}}{4}T\sum_{i=1}^{d}\frac{1}{\sqrt{T + \lambda_i(V_0)}}\Big) = \Omega\Big(T\sum_{i=1}^{d}\frac{v_i}{\sqrt{T + \lambda_i(V_0)}}\Big).$$

Then for any fix $i$, we build 4 distributions such that:

$$u^{\mathrm{on}}, \quad u_j^{(i),\mathrm{on}} = \begin{cases} -u_j^{(i),\mathrm{on}}, & j = i, \\ u_j^{(i),\mathrm{on}}, & \text{else}, \end{cases}, \quad u_j^{\mathrm{off}} = \begin{cases} -u_j^{\mathrm{on}}, & j = i, \\ u_j^{\mathrm{on}}, & \text{else}, \end{cases}, \quad u^{(i),\mathrm{off}} = u^{\mathrm{on}}.$$

Then we have:

$$P_{u^{\mathrm{on}}, u^{\mathrm{off}}}(D_0, a_{1:T}, x_{1:T}) = P_{u^{\mathrm{off}}}(D_0)\prod_{t=1}^{T}\pi_t(a_t \mid \mathcal{H}_{t-1}, D_0)\, P_{u^{\mathrm{on}}}(x_t \mid a_t),$$

$$P_{u^{(i),\mathrm{on}}, u^{(i),\mathrm{off}}}(D_0, a_{1:T}, x_{1:T}) = P_{u^{(i),\mathrm{off}}}(D_0)\prod_{t=1}^{T}\pi_t(a_t \mid \mathcal{H}_{t-1}, D_0)\, P_{u^{(i),\mathrm{on}}}(x_t \mid a_t).$$

Tips: Since they share the same history but only differ in the true parameter $\theta$, their policy is the same. Then, by analysis the KL decomposition:

$$\begin{aligned}
&KL\big(P_{u^{\mathrm{on}}, u^{\mathrm{off}}}(D_0, a_{1:T}, x_{1:T}), x_{1:T}) \,\|\, P_{u^{(i),\mathrm{on}}, u^{(i),\mathrm{off}}}(D_0, a_{1:T}, x_{1:T})\big) \\
&:= KL(P_u \| P_{u^{(i)}}) \\
&= KL(P_{u^{\mathrm{off}}}(D_0) \| P_{u^{(i),\mathrm{off}}}(D_0)) \\
&\quad + \mathbb{E}_{D_0, a_{1:T} \sim P_{u^{\mathrm{on}}}}\left[\sum_{t=1}^{T} KL\big(P_{u^{\mathrm{on}}}(x_t \mid \mathcal{H}_{t-1}, D_0) \| P_{u^{(i),\mathrm{on}}}(x_t \mid \mathcal{H}_{t-1}, D_0)\big)\right] \\
&= \frac{1}{2}\sum_{s=1}^{N}\langle\theta(u^{\mathrm{off}}) - \theta(u^{(i),\mathrm{off}}), b_s\rangle^2 + \frac{1}{2}\sum_{t=1}^{T}\langle\theta(u^{\mathrm{on}}) - \theta(u^{(i),\mathrm{on}}), a_t\rangle^2 \\
&= \frac{1}{2}\sum_{s=1}^{N}(-2\epsilon_i u_i^{\mathrm{on}} b_{s,i})^2 + \frac{1}{2}\sum_{t=1}^{T}(2\epsilon_i u_i^{\mathrm{on}} a_{t,i})^2 \\
&= \frac{v_i^2}{2(T + \lambda_i(V_0))/d}\big(\lambda_i(V_0) + T/d\big) \\
&= v_i^2/2 \\
&\leq v_{\max}^2/2 \\
&\leq S^2/2d,
\end{aligned}$$

where the last inequaltiy holds for $v_{\max} \leq S/\sqrt{d}$.

Furthermore, define event $E_t^{(i)} = \{\mathrm{sgn}(a_{t,i}) = u_i^{\mathrm{on}}\}$ and by using the Bretagnolle-Huber inequality, we have:

$$P_u\big(E_t^{(i)}\big) + P_{u^{(i)}}\big((E_t^{(i)})^c\big) \geq \frac{1}{2}\exp\big(-\mathrm{KL}(P_u \,\|\, P_{u^{(i)}})\big)$$

and sum up over t, and from $\forall a, b \in \mathbb{R}^+$, $a + b > c \Rightarrow a \geq \frac{c}{2}$ or $b \geq \frac{c}{2}$,

$$\mathbb{E}_u\big[N_i^-(u)\big] \geq \frac{T}{4}\exp(-\mathrm{KL}(P_u\,\|\,P_{u^{(i)}})) \geq \frac{T}{4}\exp\big(-S^2/2d\big)$$

Finally,

$$\mathrm{Reg}(T) = \sum_{t=1}^{T} r_t$$

$$= \frac{2}{\sqrt{d}}\sum_{i=1}^{d}\epsilon_i\,\mathbb{E}_u\big[\,N_i^-(u)\,\big]$$

$$\geq \frac{2}{\sqrt{d}}\sum_{i=1}^{d}\epsilon_i\cdot\frac{T}{4}\cdot\exp(-S^2/2d)$$

$$= \frac{e^{-\frac{S^2}{2d}}}{4}T\sum_{i=1}^{d}\frac{v_i}{\sqrt{T+\lambda_i(V_0)}}$$

$$= \Omega\left(T\sum_{i=1}^{d}\frac{v_i}{\sqrt{T+\lambda_i(V_0)}}\right).$$

**Case 2** When $\sum_i v_i \leq e^{-\frac{1}{2}(S^2/d-1)}d$, then

$$\Omega\big(\frac{e^{-S^2/2d}}{4}T\sum_{i=1}^{d}\frac{v_i}{\sqrt{T+\lambda_i(V_0)}} + \frac{e^{-1/2}}{4}T\sum_{i=1}^{d}\frac{1}{\sqrt{T+\lambda_i(V_0)}}\big) = \Omega(T\sum_{i=1}^{d}\frac{1}{\sqrt{T+\lambda_i(V_0)}}).$$

Fix $i$, we build 4 distributions such that:

$$u^{\mathrm{on}},\ u_j^{(i),\mathrm{on}} = \begin{cases} -u_j^{(i),\mathrm{on}}, & j=i, \\ u_j^{(i),\mathrm{on}}, & \text{else}, \end{cases},\ u^{\mathrm{off}} = u^{\mathrm{on}},\ u_j^{(i),\mathrm{off}} = \begin{cases} -u_j^{\mathrm{on}}, & j=i, \\ u_j^{\mathrm{on}}, & \text{else}. \end{cases}$$

Then we let $\epsilon_i = \frac{1}{2\sqrt{(T+N_i)/d}}$, and we have:

$$KL(P_u\|P_{u^{(i)}}) = KL(P_{u^{\mathrm{off}}}(D_0)\|P_{u^{(i),\mathrm{off}}}(D_0))$$

$$+ \mathbb{E}_{D_0,a_{1:T}\sim P_{u^{\mathrm{on}}}}\left[\sum_{t=1}^{T}KL\big(P_{u^{\mathrm{on}}}(x_t\mid\mathcal{H}_{t-1},D_0)\|P_{u^{(i),\mathrm{on}}}(x_t\mid\mathcal{H}_{t-1},D_0))\right]$$

$$= \frac{1}{2}\sum_{s=1}^{N}\langle\theta(u^{\mathrm{off}})-\theta(u^{(i),\mathrm{off}}),b_s\rangle^2 + \frac{1}{2}\sum_{t=1}^{T}\langle\theta(u^{\mathrm{on}})-\theta(u^{(i),\mathrm{on}}),a_t\rangle^2$$

$$= \frac{1}{2}\sum_{s=1}^{N}(-2\epsilon_i u_i^{\mathrm{on}}b_{s,i})^2 + \frac{1}{2}\sum_{t=1}^{T}(2\epsilon_i u_i^{\mathrm{on}}a_{t,i})^2$$

$$= \frac{1}{2}$$

Furthermore, define event $E_t^{(i)} = \{\mathrm{sgn}(a_{t,i})=u_i^{\mathrm{on}}\}$ and by using the Bretagnolle-Huber inequality, we have:

$$P_u\big(E_t^{(i)}\big) + P_{u^{(i)}}\big((E_t^{(i)})^c\big) \geq \frac{1}{2}\exp\big(-\mathrm{KL}(P_u\,\|\,P_{u^{(i)}})\big)$$

and sum up over $t$, and from $\forall a, b \in \mathbb{R}^+, \ a + b > c \Rightarrow a \geq \frac{c}{2}$ or $b \geq \frac{c}{2}$,

$$\mathbb{E}_u\big[N_i^-(u)\big] \geq \frac{T}{4}\exp(-\mathrm{KL}(P_u \,\|\, P_{u^{(i)}})) \geq \frac{T}{4}\exp(-1/2)$$

Finally,

$$\begin{aligned}
\mathrm{Reg}(T) &= \sum_{t=1}^{T} r_t \\
&= \frac{2}{\sqrt{d}}\sum_{i=1}^{d}\epsilon_i\,\mathbb{E}_u\big[\,N_i^-(u)\big] \\
&\geq \frac{2}{\sqrt{d}}\sum_{i=1}^{d}\frac{1}{2\sqrt{T+\lambda_i(V_0)}}\cdot\frac{T}{4}\cdot\exp(-1/2) \\
&= \Omega\left(T\sum_{i=1}^{d}\frac{1}{\sqrt{T+\lambda_i(V_0)}}\right).
\end{aligned}$$

Summarize case 1, case 2 and the discussion on pure online regret lower bound in Theorem 24.1 from Lattimore & Szepesvári (2020), we finished the proof.

$\square$

## E  EXPERIMENTS

We conduct numerical simulations to compare the performance of **hybrid-LinUCB** against two baselines: **Pure Online LinUCB** and **hybrid-LinUCB with unbiased offline data**.

The action set is defined as $\mathcal{A} = \{a \in \mathbb{R}^d : \|a\|_2 \leq 1\}, \theta_* \in \{\theta : \|\theta\|_2 \leq 1\},$, with dimension $d = 10$. The offline dataset size is varied over $N \in \{100, 1000, 5000\}$. And we set noise variance $\sigma^2 = 0.25$, and regularization parameter $\lambda = 10^{-2}$. Each experiment is repeated 50 times and we report the average cumulative regret. To construct the offline dataset, we first sample a true parameter $\theta_*$ uniformly from the unit ball, and then generate a biased version $\theta_{\mathrm{off}} = \theta_* \pm \frac{V}{\sqrt{d}}\mathbf{u}$, where $\mathbf{u}$ is a random unit vector and $V \in [0, 2]$ controls the overall bias level. For each offline dataset, we uniformly sample $N$ actions from $\mathcal{A}$, and generate the rewards according to $\theta_{\mathrm{off}}$.

For plotting, we show the mean cumulative regret and one-standard-error bands over 50 runs. All experiments are implemented in Python with NumPy, using Sherman–Morrison updates for matrix inverses.

Figure 1 reports the performance when the bias parameter $V$ is small. In this case, the performance of hybrid-LinUCB is almost indistinguishable from the unbiased variant. As $N$ increases, the benefit from offline data becomes more pronounced, leading to smaller regret overall.

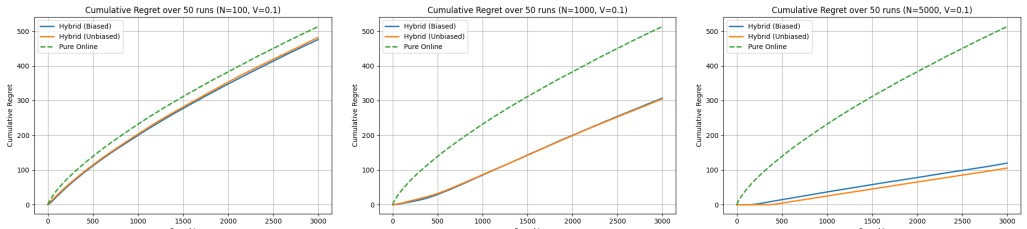

Figure 1: Performance comparison of hybrid-LinUCB with different offline data sizes when bias is small.

Figure 2 reports the performance when the bias is relatively large ($V = 0.9$, which corresponds to a per-dimension perturbation of $V/\sqrt{d} \approx 0.28$ when $d = 10$). For small $N$, the performance

of the three algorithms is similar. Despite the large bias, hybrid-LinUCB remains fairly stable. As $N$ grows, the algorithm still benefits from the additional offline data, but the improvement is less significant compared to the small-bias case. Moreover, we observe that the gap between biased hybrid-LinUCB and its unbiased counterpart widens with larger $N$, which matches our intuition: a larger offline dataset amplifies the influence of bias on online learning, thereby leading to more regret compared to the unbiased case.

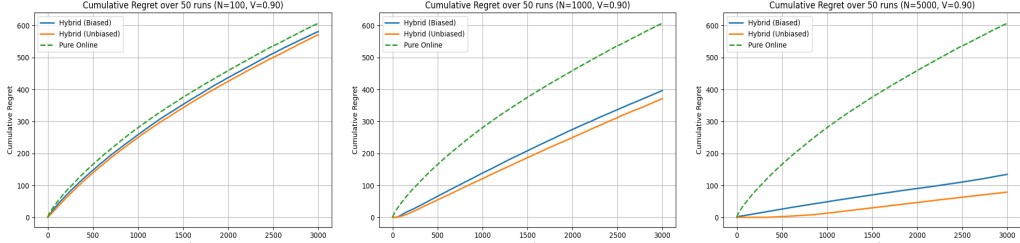

Figure 2: Performance comparison of hybrid-LinUCB with different offline data sizes when bias is large.

## USE OF LLMS

In preparing this paper, we made limited use of large language models (LLMs). They were employed only for auxiliary purposes such as improving the clarity of writing and quickly checking programming syntax or documentation. All conceptual contributions, theoretical developments, and experimental implementations are original work of the authors.

