# OpenReview forum: "Regret Analysis of Hybrid Linear Bandits with Biased Offline Data"
_ICLR.cc/2026/Conference — Submitted to ICLR 2026_

### Official Review · Reviewer_LbFP · 2025-10-29

**Soundness:** 2
**Presentation:** 2
**Contribution:** 2
**Rating:** 2
**Confidence:** 4

**Summary:**

This paper consider the hybrid bandit with an offline linear regression before the online decision making using ridge procedure. The paper proposes the hybrid LinUCB algorithm and its regret analysis.

**Strengths:**

The paper clearly state the problem and provide logical proposal (algorithms and its steps) and analysis.

**Weaknesses:**

1. Lack of experiments in main body. We don't need an intensive experiment but I think it is really important to empirically show the improvement about using the offline regression in advance. This should be the critical contribution.

2. Current problem is a little bit trivial for a paper in ICLR. I believe the most novelty is borrowing the "similar" offline regression before the online sequential making. But the investigation is shallow, including the problem formulation and development of the algorithms. See my questions for more details.

3. Some notation/writing issue such as $V$ is a constant and $V_0$ is the matrix. It is better to avoid notations like that.

**Questions:**

1. In Section 3 summary, the first bullet: in general tighter confidence set doesn't mean a better method and it is tricky to state "accelerate exploration". I would even think the proposed method is lacking in exploration ability. So this requires more investigate: which scenario needs the offline data and tighter confidence set and which ones are better to use only the online procedure. That's a critical insight for this work.

2. The overall design is close to a trivial solution: use a general $V_0$ as the initial Gram matrix and the proposed offline Gram matrix is just a particular choice for that; also use a general vector $a$ for the summation $\sum_{s=1}^t a_s x_s$ and set $a = \sum_{s=1}^N b_s y_s $ as a special case. So this means this work is better to marked as a good initial step for a project to consider the offline data for similar task to boost the online decision making. I believe it requires more effort to develop a comprehensive project.

---

### Official Review · Reviewer_qWTT · 2025-10-31

**Soundness:** 3
**Presentation:** 3
**Contribution:** 1
**Rating:** 2
**Confidence:** 3

**Summary:**

This paper studies the problem of stochastic linear bandits where there is biased offline data that has already been collected before starting the interaction with the environment. They give an algorithm that uses two confidence ellipsoids: one that uses only the online data, and one that uses both the offline and online data. The algorithm uses the union of these two ellipsoids to construct an upper confidence bound and then does UCB on this upper confidence bound. They give regret upper and lower bounds.

**Strengths:**

The paper has a clear explanation of the algorithm and analysis.

**Weaknesses:**

1. The algorithm follows almost immediately from standard techniques. In particular, it is just UCB with the construction of the confidence radius using standard least-squares confidence sets for the estimation error and then uses the minimum and maximum eigenvalue of the offline Gram matrix to bound the effects of the offline data.
2. I do not see how the regret bound shows that offline data is useful. In the best possible case of no bias ($V=0$) and $N = T$ rounds of optimal offline exploration $\lambda_{\min}(V_0) = T$, the regret bound is still $O(d \sqrt{T})$, which is the same as if there was no offline data. So the bound only shows that offline data is advantageous if there is more offline data than online data? This conclusion does not seem to suggest that offline data is particularly useful.
3. The regret bound uses the minimum and maximum eigenvalues on the offline Gram matrix, which are the loosest possible bounds, and ignores any potential correlations between offline data and the optimal action. In particular, it does not show what type of offline data is good, other than it needs to be "exploratory" and a lot $N >> T$.
4. Overall, I did not see substantial contributions over the standard linear bandit techniques or the related literature.

**Questions:**

1. Can you provide clarification on how the proposed regret bound relates to the one in Zhang et al. (2025)? It seems that the one in Zhang is often stronger. With no offline data, the regret bound in Zhang is the same as the standard setting $O(d \sqrt{T})$. With exploratory offline data $\lambda_{\min}(V_0) = T$ and no bias $V=0$, the regret bound in Zhang et al. (2025) is $\sqrt{d T}$, which is better than the regret bound in the present paper of $d \sqrt{T}$.

Some suggestions on the writing:
- I would suggest defining $V_0$ and $V$ before they are used in line 76.
- Throughout the paper, the algorithm is referred to as "intersection of confidence sets," but really it is the union of confidence sets as only one of them needs to be satisfied at a time.
- On line 214, it is not "strictly" more favorable, as it is a non-strict order $\succeq$.

---

### Official Review · Reviewer_C3q7 · 2025-11-08

**Soundness:** 3
**Presentation:** 1
**Contribution:** 2
**Rating:** 2
**Confidence:** 4

**Summary:**

This paper proposes a novel online learning algorithm, H-LinUCB, which leverages offline data to accelerate optimal action learning in the online setting. Through regret decomposition, the authors theoretically demonstrate that the algorithm performs no worse than purely online learning when the offline data is biased or has poor coverage, while achieving substantial gains when the offline data provides informative knowledge about the online environment.

**Strengths:**

The paper provides rigorous theoretical analysis, and the comparative discussion with prior work effectively clarifies the novel contributions of this study.

**Weaknesses:**

Please refer to the Questions section.

**Questions:**

1. Some typos affect the readability and the understanding of the paper. For example, in the main paper, the reviewer did not find the mathematical definition of V0, except one line definition, “offline feature matrix V0” at line 19 in the abstract. Also, at Lemma 1, equation (2), does $\sqrt{\beta_N}$ have the same definition rule of $\sqrt{t}$? I think it is better to specify the range of $t$ to make the definition clearer.

2. Restrictiveness of the linearity assumption. Is the proof extendable to generalized linear bandits with a known link function $g$?

3. Lacks of discussion of computational complexity of the current algorithm. The reviewer suggests a comparison of computational complexity of H-LinUCB to the existing literature.

4. Lack of discussion of the benefit of H-LinUCB. Even though the author mentions H-LinUCB’s benefit at line 243-251, however, it is still not clear, for example, what the highest possible benefit H-LinUCB could be. For example, in the extreme case, if $V$ = 0, the offline data is iid, $\lambda_{min}(V_0)$ grows linearly with $N$ [1], then if $N = \Omega(T^2)$, H-LinUCB can achieve linear regret. Or another discussion when $V$ is small, in polynomial order of $\frac{1}{N}$, how the upper bound could be. The reviewer believes that this discussion will better highlight the contribution of this paper.

5. More comprehensive simulations are recommended. It will be more convincing to conduct simulations not only for H-LinUCB, but also for other algorithms discussed in Section 4 for comparison. Also, experiments on real-world datasets are strongly recommended. There are many available real-world datasets / real-world datasets simulators (trained by diffusion models which have the similar distribution of the real-world data). The reviewer will significantly improve the rating if the author conducts the large-scale real-world experiments. The author will reconsider the rating if all the questions are properly addressed.

References:

[1] Provably Optimal Algorithms for Generalized Linear Contextual Bandits

---

### Official Review · Reviewer_MLBJ · 2025-11-10

**Soundness:** 3
**Presentation:** 4
**Contribution:** 3
**Rating:** 4
**Confidence:** 4

**Summary:**

This paper considers the problem of learning linear bandits in which the linear model has shifted since the last time data was collected from it. More explicitly, the model consists of 2 phases. In phase 1, a policy pulls arms in a linear bandit whose 'offline' parameter is $\theta^{\text{off}}$ and gathers data in a dataset. In phase 2, an agent is given this dataset and faces the same linear bandit but with an 'online' parameter being $\theta_* \neq \theta^{\text{off}}$ instead of $\theta^{\text{off}}$.

Under the assumption that the amount of drift between the offline and online $\theta$ parameters is at most a known upper bound, the paper develops an algorithm to utilize the offline dataset to 'warm start' an online bandit algorithm and maintain valid confidence set bracketing of the online parameter. It shows that this property can be used to bound the regret during online operation by a quantity that captures the effect of the initial bias of the offline dataset, as well as its advantage in warm-starting the online procedure if the drift is low. The paper's results are purely theoretical, and include an impossibility result (fundamental lower bound on regret) showing that the nature of the dependence of the regret bound on the spectrum of the offline data Gram matrix cannot be circumvented.

**Strengths:**

The paper is strong in its consideration of a practical decision making problem: bandits with structured (feature-based) arms in a parametric model where the parameter undergoes a shift from one application run to another. I appreciate the clean and transparent model formulation involving two parameters - one offline and the other online.

The solution approach followed in the paper, while based on the well-known statistical decision making principle of optimism under uncertainty within a confidence set, is creative in its use of combining two known data sources (the offline data set and the online data) together with a structural coupling between them (a bounded drift) to get tight, anytime confidence sets for the online parameter, which is used to guide arm playing.

The paper also goes to great lengths to situate its approach and results against the backdrop of other recent pieces of related work. The thorough attention to detail in comparing and contrasting its contribution with other work is commendable.

**Weaknesses:**

- It is not very satisfying for the paper's modeling approach to assume a known upper bound on the (L2 norm in this paper) deviation, and develop an algorithm that maintains a high-probability confidence set for the true online parameter *only when the upper bound holds*. In other words, the algorithm design and associated guarantee does not have a natural 'continuity' property with respect to the problem instance -- when, for instance, the norm difference between the online and offline theta parameters is just slightly over the assumed bound $V$ in reality, all bets are off in a theoretical sense. A more graceful and robust approach to the online-offline biased linear bandit problem would instead develop regret bounds that are adapted to the true (unknown) deviation, using an algorithm that does not need the additional hyperparameter $V$.

- A secondary consequence of the above, which is not ideal, is that the obtained regret bound (in Theorem 1) does not actually capture the true deviation or geometry of theta_offline relative to theta_online. Yes, I do notice that the bound insightfully captures the notion of coverage by way of the spectrum of the offline Gram matrix $V_0$; however, it depends only on the worst deviation $V$. If, say, there is no actual deviation but the algorithm still uses a large bound hyperparameter $V \gg 0$, the regret bound is not refined enough to capture the (intuitively) small regret that the algorithm could potentially enjoy.

- While I appreciate the attention to detail in comparing and explain the paper's result with earlier work, I think that an unduly large amount of space has been devoted to this task (just over a page's worth). Again, the proof sketch for Theorem 2 appears to be relying on rather standard information-theoretic tools (confusing instances, change of measure identities, Bretagnolle-Huber, etc.) and is also unwarranted to an extent. This space could have been used more effectively to demonstrate extensive numerical experiments benchmarking the proposed approach with closely related approaches (that the paper argues do not exhibit strong regret bounds). While I see that experiments are reported in the appendix, I do not think the paper would be making a strong case if no experimental data were reported in its main body.

**Questions:**

1. Could the author(s) please comment on the feasibility of designing an algorithm without needing the knowledge of the the additional deviation hyperparameter $V$, and the possibility of obtaining regret guarantees that instead degrade gracefully with the amount of true deviation?

2. The choice of confidence ellipsoids that are built to be intersected in Algorithm 1 appears to be a rather specific and deliberate. There could be alternative prescriptions, e.g., why not adopt a more 'natural' approach that takes the purely offline data ellipsoid, inflates it using the max. deviation hyperparameter $V$, and intersects it with the purely online data ellipsoid, instead of intersecting the purely online and a hybrid ellipsoid? Could the author(s) shed light on this technical design choice?

---

### Meta-Review · Area_Chair_UPcH · 2026-01-05

**Summary:**

This paper works on a hybrid linear bandits problem, where an offline dataset could be used to improve the online learning performance. The reviewers reached a consensus that the paper lacks technical novelty and relies on overly restrictive assumptions. Therefore, my recommendation is to reject,

**Reviewer Concerns:**

The authors did not present responses. All the concerns are left unresolved.

**Reviewer Scores:**

There is no response, so the reviewers would have maintained their original scores had they been able to engage fully in the discussion.

---

### Decision · Program_Chairs · 2026-01-26

Reject